# MCL-1 Inhibitor S63845 Distinctively Affects Intramedullary and Extramedullary Hematopoiesis

**DOI:** 10.3390/pharmaceutics15041085

**Published:** 2023-03-28

**Authors:** Hexiao Zhang, Fei Li, Ming Yang, Wenshan Zhang, Mei He, Hui Xu, Chaoqun Wang, Yiran Zhang, Wei Wang, Yingdai Gao, Xue Du, Yinghui Li

**Affiliations:** 1State Key Laboratory of Experimental Hematology, National Clinical Research Center for Blood Diseases, Haihe Laboratory of Cell Ecosystem, PUMC Department of Stem Cell and Regenerative Medicine, CAMS Key Laboratory of Gene Therapy for Blood Diseases, Institute of Hematology & Blood Diseases Hospital, Chinese Academy of Medical Sciences & Peking Union Medical College, Tianjin 300020, Chinaydgao@ihcams.ac.cn (Y.G.); 2Tianjin Institutes of Health Science, Tianjin 301600, China; 3Department of Gynecology and Obstetrics, Tianjin Medical University General Hospital, Tianjin 300052, China; 4Department of Gynecology, Tianjin Union Medical Center, Tianjin Medical University, Tianjin 300121, China

**Keywords:** targeted drugs, MCL-1 inhibitor, hematopoietic suppression, extramedullary hematopoiesis

## Abstract

Conventional chemotherapy for killing cancer cells using cytotoxic drugs suffers from low selectivity, significant toxicity, and a narrow therapeutic index. Hyper-specific targeted drugs achieve precise destruction of tumors by inhibiting molecular pathways that are critical to tumor growth. Myeloid cell leukemia 1 (MCL-1), an important pro-survival protein in the BCL-2 family, is a promising antitumor target. In this study, we chose to investigate the effects of S63845, a small-molecule inhibitor that targets MCL-1, on the normal hematopoietic system. A mouse model of hematopoietic injury was constructed, and the effects of the inhibitor on the hematopoietic system of mice were evaluated via routine blood tests and flow cytometry. The results showed that S63845 affected the hematopoiesis of various lineages in the early stage of action, causing extramedullary compensatory hematopoiesis in the myeloid and megakaryocytic lineages. The maturation of the erythroid lineage in the intramedullary and extramedullary segments was blocked to varying degrees, and both the intramedullary and extramedullary lymphoid lineages were inhibited. This study provides a complete description of the effects of MCL-1 inhibitor on the intramedullary and extramedullary hematopoietic lineages, which is important for the selection of combinations of antitumor drugs and the prevention of adverse hematopoiesis-related effects.

## 1. Introduction

The conventional cancer treatment approach of chemotherapy, in which cancer cells are eradicated by cytotoxic drugs, is a crucial component of the treatment of a wide variety of cancers [1]. Chemotherapeutic drugs are efficient at destroying malignant cells; however, these drugs’ limited therapeutic index, low selectivity, and high toxicity make them less effective than they could be [2]. The improved understanding of the malignant mechanisms that produce the disease has resulted in the development of innovative therapeutic alternatives. One of them is the manufacture of extremely accurate, small-molecule, targeted pharmaceuticals. For example, in recent years, several studies have focused on the synthesis of effective small-molecule compounds, known as tubulin inhibitors, in the search for potent and selective antineoplastic agents [3,4,5]; meanwhile, some small-molecule inhibitors that target FLT3 have been intensively studied and could be effective modalities for the treatment of acute myeloid leukemia [6]. Targeted treatments achieve efficient and targeted elimination of cancer by inhibiting the molecular pathways that are crucial for tumor development [7]. Targeted molecular therapy possesses significant advantages over traditional chemotherapy. On the one hand, single-agent chemotherapy typically induces drug resistance to several agents [8]. Targeted medicines destroy cancer cells through a variety of therapeutic pathways, and their combination with conventional chemotherapeutic drugs can enhance their therapeutic efficacy [9]. In a phase III clinical trial (NCT02993523), 65% of elderly AML patients who received first-line treatment with Venetoclax, in conjunction with cytarabine, achieved complete remission or complete remission with incomplete hematologic recovery [10]. On the other hand, targeted therapies are theoretically less harmful than cytotoxic medications. Chemotherapy is centered on preventing the division of rapidly proliferating cells and suppresses cancer cells while impacting rapidly proliferating healthy cells, resulting in a variety of side effects. By inhibiting certain signaling pathways, targeted therapy reduces undesirable effects [11]. However, the damage caused to the hematopoietic system by targeted agents, and the pattern of the hematopoietic system’s response to such agents, remain unknown and require additional research.

Apoptosis is an evolutionarily conserved process of programmed cell death and is essential for the elimination of unwanted or potentially dangerous cells [12]. The BCL-2 family of proteins is a crucial component of the mitochondrial apoptotic pathway, which consists of a pro-survival group and two subgroups of pro-apoptotic cell death effectors; these are called multi-BH structural domain cell death effectors and BH3-only cell death effectors [13]. The effective clinical testing of BCL-2-targeting small-molecule inhibitors demonstrates that BH3 analogs can cause apoptosis in malignant cells [14]. In preclinical models of lymphoma and small-cell lung cancer, the first BH3 mimic, ABT-737, exhibited anticancer efficacy as a single drug [15]. In a phase I clinical trial (NCT00406809), it was demonstrated that Navitoclax, a second-generation BH3 analog, is effective against B-cell cancers [16]. Venetoclax, a mono-selective BCL-2 inhibitor [17], was utilized in a phase II clinical investigation in patients who had CLL with chromosome 17p deletion (NCT01889186) [18] and was later licensed for clinical use by the U.S. Food and Drug Administration (FDA) [19]. MCL-1 plays a crucial role in the process of guaranteeing cell survival and is usually shown to be genetically elevated in human cancers [20]. Certain medications that inhibit gene transcription in cancer cells exert their deadly effects on other cells, in part, by downregulating MCL-1 [21]. S63845 is a small-molecule inhibitor that targets MCL-1 with high selectivity and potency; hence, it is considered a very promising candidate for use as a targeting agent [22]. However, the effects of medications that target the BCL-2 pathway on regular hematopoiesis, and the response of the normal hematopoietic system to this class of targeted therapy, are unknown and require additional research.

Hematopoiesis is the process of the formation of blood cell components and refers to the formation, growth, maturation, and specialization of blood cells from hematopoietic stem cells to hematopoietic progenitor cells and then to specialized blood cells [23]. The bone marrow (BM) is the primary site of adult hematopoiesis and hosts the majority of all hematopoietic stem cells and progenitor cells [24]. Long-term hematopoietic stem cells located at the apex of the structural hierarchy gradually differentiate in the bone marrow to produce mature blood cells, including myeloid cells (My), megakaryocytes (Mk), erythroid cells (Er), and lymphocytes (Ly) [25]. Extramedullary hematopoiesis (EMH) is described as the production of mature blood cells outside the bone marrow cavity and is secondary to a pathological state of bone marrow insufficiency or ineffective hematopoiesis [26]. Chemotherapy for cancer treatment damages the hematopoietic system, reducing the number of hematopoietic stem cells in the bone marrow and peripheral circulation. This causes anemia and immunodeficiency [27]. Neutropenia is a common side effect of anticancer therapy and may increase patients’ susceptibility to bacterial or viral infections [28]. Adverse effects, such as neutropenia and thrombocytopenia, have also been reported in patients based on the recent clinical use of Venetoclax [29]. However, the effects of specific small-molecule inhibitors on the body’s hematopoietic system are poorly understood, such as the effect of the drug on the differentiation and maturation of all lineages of hematopoietic cells and the sensitivity of intramedullary and extramedullary hematopoiesis to drug stimulation, all of which need to be addressed through the establishment of an appropriate animal model.

In the present study, we designed a mouse model of targeted drug-induced hematopoietic injury, in which S63845, a small-molecule targeted drug, was injected into the tail veins of healthy CB6F1 mice for five consecutive days to study the effects of targeted drugs on hematopoiesis in the organism. The MCL-1 inhibitor S63845 was found to have some effects on all lineages of normal hematopoiesis in mice, especially in the early period. Myelopoiesis, megakaryopoiesis, and erythropoiesis mainly affected extramedullary compensatory hematopoiesis, with a higher proportion of relatively mature myeloid and differentiated Mk-lineage cells, while erythropoietic cells were mostly blocked in the early stage of the differentiation of progenitor cells; the effects on lymphoid hematopoiesis were inhibited in both intramedullary and extramedullary hematopoiesis. These findings have implications for the clinical selection of combination antitumor drug therapy regimens and for the prediction and prevention of adverse hematopoiesis-related effects.

## 2. Materials and Methods

### 2.1. Compounds

MCL-1 inhibitor S63845 was purchased from MedChemExpress (HY-100741) and stored in the dark. S63845 was formulated extemporaneously in 25 mM HCl and 20% 2-hydroxy propyl β -cyclodextrin (Solarbio).

### 2.2. Mice

Female CB6F1 mice (6–8 weeks old) were purchased from BEIJING HFK BIOSCIENCE CO., LTD. All of the mice were kept under specific-pathogen-free (SPF) conditions, with free access to food and water. All of the experimental procedures using mice were performed in accordance with the Regulations for the Administration of Affairs Concerning Experimental Animals, approved by the State Council of the People’s Republic of China. At the end of all experiments, the animals were euthanized under CO_2_ anesthesia. The animal use protocol listed below has been reviewed and approved by the Animal Ethical and Welfare Committee (AEWC) of the Institute of Hematology & Blood Diseases Hospital (IHCAMS-DWLL-NKRDP2022007-1).

### 2.3. Hematopoietic Suppression Model

All female mice (6–8 weeks old) were randomly and evenly classified, by weight, into different experimental groups after one week of adaptive rearing under SPF conditions. Mice were classified into three groups: the control group was treated with a vehicle, and the 25 mg/kg and 50 mg/kg groups were injected with the S63845 solution. S63845 was dissolved in 25 mM HCl and 20% 2-hydroxy propyl β-cyclodextrin, which were used as soon as they were ready. The mice were treated with a dose of vehicle or S63845 (the 25 mg/kg and 50 mg/kg groups) via tail vein injection. This was administered once a day for 5 consecutive days from day 0 (the first day of administration) to day 4. The mice were sacrificed on day 7 or day 22 in preparation for the follow-up experiments. 

### 2.4. Complete Blood Counts (CBCs)

Peripheral blood (PB) was collected from the tail veins of living mice and analyzed using an automated hematology analyzer (Sysmex, XN-9000) according to the manufacturer’s instructions.

### 2.5. Complete Bone Marrow Counts

Mouse BM cells were flushed out from their ilia, femurs, and tibias into phosphate-buffered saline (PBS) with 2 mM ethylenediaminetetraacetic acid (EDTA, Sigma-Aldrich, St. Louis, MO, USA) in preparation for the follow-up experiments.

### 2.6. Spleen (SP) Index Measurement

The spleens of mice from each group were removed and weighed on an electronic analytical balance after blotting excess water using filter paper, and their SP indexes were calculated (SP index = SP weight, mg/body weight, 10 g).

### 2.7. Wright–Giemsa Staining and Hematoxylin–Eosin (HE) Staining

BM cytospins were used, with the slides prepared in a cytocentrifuge (Cytospin 4, Thermo Scientific, Waltham, MA, USA) at 400 rpm for 5 min, followed by Wright–Giemsa staining (G1020, Solarbio Science & Technology, Beijing, China). The brightfield slides were scanned using a NanoZoomer S360 (Hamamastu Photonics, Hamamatsu, Japan) at 400× magnification. The images were recorded and analyzed using NDP. view 2.9.22 RUO. The paraformaldehyde-fixed spleen tissues were embedded in paraffin and cut in preparation for HE staining.

### 2.8. Flow Cytometry Analysis

Mouse BM cells were flushed out from the ilia, femurs, and tibias as previously described, and the spleens were separated into two parts; one half was ground for use in flow cytometry analysis, and the other half was used for HE staining. In addition, peripheral blood cells were obtained from the tail vein. Cells were stained at 4 °C for 30–60 min in PBS with a combination of the following antibodies and fluorophores: a mixture of lineage-specific antibodies (PE-Cy7-labeled anti-mouse CD3e, CD4, CD8, B220, Gr-1, Mac-1, and Ter-119 (Invitrogen, Waltham, MA, USA; 25-0031-81, 25-0041-82, 25-0081-82, 25-0452-81, 25-5931-82, 25-0112-81, and 25-5921-82)), PerCP-Cy5.5-labeled anti-mouse Sca-1 (Invitrogen; 45-5981-82), APC-labeled anti-mouse c-Kit (Invitrogen; 17-1171-83), FITC-labeled anti-mouse CD34 (Invitrogen; 11-0341-85), BV421-labeled anti-mouse CD16/32 (Biolegend, San Diego, CA, USA; 101332), PE-labeled anti-mouse Flk-2 (Invitrogen; 12-1351-82), APC-Cy7-labeled anti-mouse IL7Rα (Biolegend; 135040), FITC-labeled anti-mouse CD45 (BD, East Rutherford, NJ, USA; 553080), BV510-labeled anti-mouse CD41 (Biolegend; 133923), PerCP-Cy5.5-labeled anti-mouse CD42d (Biolegend; 148508), APC-labeled anti-mouse CD71 (Biolegend; 113819), PE-labeled anti-mouse CD3e (Invitrogen; 12-0031-83), PerCP-Cy5.5-labeled anti-mouse B220 (Invitrogen; 45-0452-82), APC-labeled anti-mouse Mac-1 (Invitrogen; 17-0112-83), APC-Cy7-labeled anti-mouse Gr-1 (Biolegend; 108424), and PE-Cy7-labeled anti-mouse NK1.1 (Invitrogen; 25-5941-82). Following a washing step, stained cells were analyzed using an FACS CantoII (BD) flow cytometer. Data were analyzed using FlowJo V10 software.

### 2.9. Statistical Analysis

All data were presented as the mean ± standard error of the mean (SEM) from at least three independent biological replicates. Statistical analyses were performed using GraphPad Prism software version 8.0. Statistical differences were evaluated using a two-tailed Student’s *t*-test, with significance at *p*-values < 0.05. (* *p* < 0.05, ** *p* < 0.01, *** *p* < 0.001, **** *p* < 0.0001, and ns = no significance).

## 3. Results

### 3.1. S63845 Affects Extramedullary Hematopoiesis in Its Early Stage after Treatment 

To test the effect of targeted drugs on the hematopoietic system, we designed a mouse model in which a solvent control and various doses of S63845 (25, 50 mg/kg) were injected into the tail veins of healthy CB6F1 mice for five consecutive days and were analyzed on day 7 and 22 after treatment (Figure 1A). We observed no significant weight loss in the S63845-treated groups (Figure 1B). Complete blood counts were performed to assess peripheral blood cell changes on days 7, 10, 13, 16, 19, and 22. On day 7, we observed a significant decrease in red blood cells (RBCs) in both the low- and high-dose S63845-treated groups compared to those in the control group; then, the RBCs returned to normal on day 22. For platelets (PLTs), both of the S63845-treated groups showed an increase on day 7 relative to the control group. Moreover, on day 22, the PLT count in the high-dose group still showed an increase compared with the control group, whereas they recovered in the low-dose group. The white blood cell (WBC) count was generally in a state of fluctuation in each group and showed no significant difference compared to the control group on day 22. Compared with the control group, the percentage of lymphocyte (LYMPH) was significantly reduced in the high-dose group on day 7 and day 22, while that in the low-dose group was not significantly different from that in the control group at either time point. On the contrary, the percentage of neutrophils (NEUTs) in the high-dose group was higher than that in the control group, and there was no significant difference between that in the low-dose group and the control group on day 7 and day 22 (Figure 1C). These data suggest that after the administration of S63845, erythropoiesis and lymphopoiesis were suppressed in the early stage and could be recovered in the late stage. An increase in the PLT count was observed throughout the observation period. 

In order to detect whether intramedullary or extramedullary hematopoiesis is more sensitive to drug stimulation, we observed the condition of the BM and SP. Additionally, the BM morphology and complete BM counts demonstrated that there was not much difference among the three groups on day 7 and day 22 (Figure 1D). S63845 treatment had no significant effect on the morphology of the SP, as indicated by HE staining (Figure 1E). However, we measured the SP index, which showed a significant increase in both treatment groups on day 7 compared to the control group and recovered on day 22 (Figure 1E). These data indicate that the targeted drug had a greater impact on extramedullary hematopoiesis in the early stage and could be recovered in the later stage.

### 3.2. S63845 Treatment Mobilized the Differentiation of Hematopoietic Stem and Progenitor Cells (HSPCs)

To identify the effect of the targeted drug on HSPCs, we used flow cytometry analysis to compare differences in the percentages of long-term hematopoietic stem cells (LT-HSCs), short-term hematopoietic stem cells (ST-HSCs), and multipotent progenitors (MPPs) between the control group and treatment groups. It is well known that CD34 and Flk-2 markers are usually used in combination with Lin^−^c-Kit^+^Sca-1^+^ (LKS^+^) to sort LT-HSCs (CD34^−^Flk-2^−^LKS^+^), ST-HSCs (CD34^+^Flk-2^−^LKS^+^), and MPPs (CD34^+^Flk-2^+^LKS^+^) [30,31] (Figure 2A,C). On day 7, in the BM, LT-HSCs in the low-dose group were significantly increased compared with the control group, while ST-HSCs in the low-dose group were reduced compared to the control group. However, there was no difference in MPPs among each group. On day 7, in the SP, LT-HSCs in the high-dose group were significantly increased compared with the control group. ST-HSCs in the low-dose group were reduced compared to the control group. MPPs were inhibited in both the high- and low-dose groups relative to the control group. These results indicate that the targeted drug mainly affected the MPP proportion of extramedullary hematopoiesis in the early stage (Figure 2B).

On day 22, in the BM, LT-HSCs in the high-dose group were significantly increased compared with the control group. ST-HSCs in the low-dose group were still at a lower level compared to the control group. Additionally, MPPs in the high-dose group were decreased compared with the control group. On day 22, in the SP, LT-HSCs in the high-dose group were slightly increased relative to the control group, while ST-HSCs and MPPs showed no differences among the three groups (Figure 2D). We found that there was still an effect of S63845 on late intramedullary hematopoiesis, especially LT-HSCs in the high-dose group, which were higher than that in the control group, while extramedullary hematopoiesis in the treatment group was basically the same as that in the control group. Taken together, these results indicate that S63845 treatment mobilized the differentiation of HSPCs.

### 3.3. S63845 Affects Extramedullary Myeloid Hematopoiesis in Mice

It was reported that downstream MPPs mainly differentiated into lymphoid and myeloid lineages, including common myeloid progenitors (CMPs) and common lymphoid progenitors (CLPs); CMPs differentiated into granulocyte–monocyte progenitors (GMPs) and megakaryocyte-erythroid progenitors (MEPs); then, they differentiated into downstream mononuclear progenitor cells, macrophage precursors, megakaryocyte progenitor cells, and erythroid progenitor cells, and eventually developed into mature functional cells of each lineage [32]. It is well known that CD34 and CD16/32 markers are usually used in combination with Lin^−^c-Kit^+^Sca-1^−^ (LKS^−^) to sort CMPs, GMPs, and MEPs [33]. Mac-1^+^Gr-1^+^ was used as a surface marker for mature myeloid cells. Next, we assessed the effect of S63845 on myeloid cell differentiation. In intramedullary hematopoiesis, the level of CMPs was significantly suppressed in the treatment groups compared with the control group on day 7, while the percentage of GMPs and Mac-1^+^Gr-1^+^ cells showed no significant difference (Figure 3A). The inhibitory effect of CMPs was restored on day 22, and there was no significant difference between the treatment groups and the control group. The high-dose group underwent a significant increase in the percentage of Mac-1^+^Gr-1^+^ cells at this stage (Figure 3B). Therefore, in intramedullary hematopoiesis, CMPs were inhibited in the initial stage and recovered in the later stage.

In the SP, on day 7, CMPs and GMPs were significantly increased in both treatment groups, and the effect occurred in a dose-dependent manner (Figure 3C). The levels of CMPs and GMPs in the low-dose group returned to the control level on day 22, while those in the high-dose group remained significantly higher than those in the control group (Figure 3D). In addition, there was no significant difference in the levels of Mac-1^+^Gr-1^+^ cells among the groups on day 7, and by day 22, the high-dose group showed a much higher level than the control group (Figure 3C,D). Based on the above results, we concluded that in extramedullary hematopoiesis, CMP was mobilized in the early stage and recovered in the late stage.

Remarkably, in the peripheral blood, the changes in the levels of Mac-1^+^Gr-1^+^ cells were different. The proportion of Mac-1^+^Gr-1^+^ cells in the high-dose group was higher than that in the control group in the early stage, but there was no significant difference among the groups in the late stage (Figure 3E,F). Thus, mature myeloid cells in the peripheral blood increased in the initial stage and recovered later.

### 3.4. S63845 Activates Extramedullary Megakaryopoiesis in the Early Stage and Intramedullary Megakaryopoiesis in the Later Stage

MEPs have previously been isolated based on the surface makers of CD34 and CD16/32 in combination with LKS^−^ [33]. The presence of CD45^+^CD41^−^, CD45^−^CD41^+^, CD45^−^CD41^+^CD42d^+^, and CD45^−^CD42d^+^ subpopulations indicate increasingly mature megakaryocytes [34,35,36]. For intramedullary hematopoiesis in the early stage, there was no significant difference among the other subpopulations, except for MEPs, whose levels were high in the low-dose group (Figure 4A). Moreover, for extramedullary hematopoiesis in the early stage, MEPs in both treatment groups were significantly higher than those in the control group, intermediate-stage subsets in both treatment groups were lower, and relatively mature CD45^−^CD42d^+^ subset in both treatment groups was significantly higher (Figure 4B).

For intramedullary hematopoiesis in the late stage, differences in the MEP and relatively mature CD45^−^CD42d^+^ subpopulations in both treatment groups had no statistical significance compared to the control group. Moreover, the percentages of intermediate-stage subpopulations, such as CD45^+^CD41^−^ in both the high- and low-dose groups, CD45^−^CD41^+^ in the high-dose group, and CD45^−^CD41^+^CD42d^+^ in the low-dose group, were higher than those of the control group (Figure 4C). However, there was no obvious difference among the percentages of intermediate-stage subpopulations in these three groups in extramedullary hematopoiesis (Figure 4D).

Regarding the subpopulations in the peripheral blood, there was no noticeable change among the groups on day 7 and day 22 (Figure 4E,F). As previously mentioned, the PLT count was increased in the treatment group compared with the control group (Figure 1C). These data indicate that early-onset extramedullary megakaryopoiesis activation and late-onset intramedullary hematopoietic activation maintain high levels of PLTs.

### 3.5. S63845 Affects Erythropoiesis in the Early Stage

The presence of CD45^+^CD71^+^, CD45^−^CD71^+^, CD45^−^CD71^+^Ter119^+^, and CD45^−^Ter119^+^ subpopulations indicates that erythroid cells have gradually differentiated and are mature [37,38]. On day 7, in intramedullary hematopoiesis, we found a greater increase in CD45^+^CD71^+^ cells in the high-dose group compared to the control group. The percentage of CD45^−^CD71^+^ and CD45^−^CD71^+^Ter119^+^ cells was increased in both treatment groups and was significantly different between the low-dose group and the control group. However, CD45^−^Ter119^+^ cells decreased in the treatment groups, and their levels were significantly different between the low-dose group and the control group (Figure 5A). On day 7, in extramedullary hematopoiesis, the percentage of CD45^+^CD71^+^, CD45^−^CD71^+^, and CD45^−^CD71^+^Ter119^+^ cells was significantly increased in both treatment groups relative to the control group, while the levels of CD45^−^Ter119^+^ cells were almost the same in the three groups, with no statistical significance (Figure 5B). In terms of fold change, changes in the subpopulations of extramedullary hematopoiesis were greater compared with those of intramedullary hematopoiesis (Figure 5A,B). In the peripheral blood, CD45^−^CD71^+^Ter119^+^ cell levels in the treated groups were higher than those in the control group on day 7, while CD45^−^Ter119^+^ cells were decreased in the treatment groups compared with the control group (Figure 5C). Interestingly, on day 22, whether in the BM, SP, or PB, the erythroid lineage cells in the treatment groups returned to their control group levels, with no significant differences compared with the control group (Figure 5D–F). Combining the results of the routine blood RBC count (Figure 1C) and MEP subpopulations, S63845 reduced the early RBC count and significantly inhibited the levels of intramedullary CD45^−^Ter119^+^ cells; moreover, the early differentiated progenitor cells of this lineage play a compensatory role in extramedullary hematopoiesis.

### 3.6. S63845 Inhibits Both Intramedullary and Extramedullary Lymphoid Hematopoiesis in the Early Stage

To assess the effect of S63845 on the lymphoid lineage, we analyzed the ratio of intramedullary and extramedullary CLP, CD3e^+^ (T-lymphocyte subpopulation), B220^+^ (B-lymphocyte subpopulation), and NK1.1^+^ (NK-lymphocyte subpopulation) subpopulations. CLPs mainly differentiated into B-lineage progenitor cells, T-lineage progenitor cells, some NK progenitor cells, and dendritic cells and eventually formed mature terminally differentiated cells of each lineage [39]. A CLP subset was previously isolated based on the surface makers of IL7Rα and Flk-2 in combination with Lin^−^c-Kit^Low^Sca-1^Low^ (LKS^Low^) [39,40], and CD3e^+^, B220^+^, and NK1.1^+^ subsets represented T, B, and NK differentiated mature cells, respectively. 

The differences in the percentages of CLPs in BM showed no significant difference among the three groups on day 7. However, the proportions of CD3e^+^, B220^+^, and NK1.1^+^ cells in the treated group were all significantly inhibited on day 7 in BM compared with the control group (Figure 6A). On day 22, in the BM, CD3e^+^ and NK1.1^+^ cells in both treatment groups were returned to their control group levels, while the proportion of CLPs decreased in the high-dose group relative to the control group. Furthermore, B220^+^ cells were still reduced in both the 25 mg/kg and 50 mg/kg groups on day 22 (Figure 6B).

Notably, in extramedullary hematopoiesis, all differentiated lymphoid cells, including CLP, CD3e^+^, B220^+^, and NK1.1^+^ cells, were markedly reduced in the treated groups on day 7 compared to the control group (Figure 6C). On day 22, most of them had returned to their control group levels, except for B220^+^ and NK1.1^+^ cells, whose levels remained lower in the high-dose group compared to the control group (Figure 6D).

We also evaluated lymphocytes in the peripheral blood via flow cytometry analysis. In both treatment groups, CD3e^+^ cells were decreased on day 7 compared with the control group. Additionally, B220^+^ cells were reduced in the high-dose group compared with the control group (Figure 6E). Moreover, all the lymphocytes that we measured in the treated groups returned to normal on day 22, and there were no statistically significant differences with the control group (Figure 6F). In summary, our experiments indicate that S63845 inhibits both intramedullary and extramedullary lymphoid hematopoiesis in the early stage.

## 4. Discussion

In recent years, small-molecule inhibitors have gradually become a major therapeutic class in oncology treatment due to their specific killing of cancer cells [41]. Small-molecule inhibitors are better absorbed by the body than monoclonal antibodies and can act more extensively on extracellular and intracellular targets [11]. To date, the FDA has approved 43 small-molecule inhibitors for oncology purposes, and many of these drugs have shown superiority over cytotoxic chemotherapy. However, the short- and long-term intramedullary and extramedullary effects of small-molecule inhibitors on the body’s ability to perform hematopoiesis during administration are unclear, and the elucidation of these issues is critical for the clinical application of targeted small-molecule agents. BCL-2 family inhibitors are widely used in the clinical treatment of malignant hematological tumors; they function by controlling pro- and anti-apoptotic intracellular signaling and are representative of small-molecule inhibitor-targeted drugs [42]. We selected S63845, a specific inhibitor of pro-survival protein MCL-1 in the BCL-2 family [22], and used a CB6F1 hematopoietic injury mouse model to elucidate the alterations that occur in various lineages in the hematopoietic system of the body under the stimulation of targeted drugs, including the short- and long-term effects of drug action and changes in intramedullary and extramedullary hematopoiesis. 

In this study, targeted small-molecule inhibitors were found to affect normal hematopoiesis in all lineages. Myelopoiesis, megakaryopoiesis, erythropoiesis, and lymphopoiesis were affected differently in terms of the temporal dimension of early and late drug action. In stem progenitor cells, early extramedullary MPPs were suppressed, and LT-HSCs were stressfully elevated; later, the extramedullary effects were largely restored, but at this time, the drug inhibition of intramedullary MPPs became apparent. In myeloid hematopoiesis, early intramedullary CMPs were suppressed, while extramedullary CMPs were activated by stress; later, intramedullary CMPs were restored, but the extramedullary high-dose group remained activated. In addition, both intramedullary and extramedullary Mac-1^+^Gr-1^+^ myeloid mature cell levels were later increased. This finding is consistent with existing research in which it was found that BCL-2 was not essential for the survival of mature monocytes and granulocytes, as these cell types were resistant to in vitro treatment with the BCL-2 inhibitor Venetoclax [43]. For megakaryopoiesis, early differentiated extramedullary megakaryocytes (CD45^+^CD41^+^ and CD45^−^CD41^+^) were suppressed, while relatively mature differentiated cells (CD45^−^CD42d^+^) were elevated; both recovered in the later stages, while early differentiated intramedullary megakaryocytes (CD45^+^CD41^+^, CD45^−^ CD41^+^, and CD45^−^CD41^+^CD42d^+^) were activated. In addition, in recent research, the effect of BCL-2 on platelet production and platelet lifespan was found to be dispensable, and the combined loss of BCL-2 and MCL-1 did not affect platelet production or platelet survival [44]. In erythroid hematopoiesis, the early intramedullary and early extramedullary differentiation of erythroid progenitors (CD45^+^CD71^+^, CD45^−^CD71^+^, and CD45^−^CD71^+^Ter119^+^) was activated, especially in extramedullary areas; they all recovered later. BCL-X is a member of the BCL-2 family and is thought to be important for the survival and maturation of various cell types in the erythroid cell lineage. It has also been shown that BCL-X is only necessary for the survival of mature end-stage erythrocytes [45]. In addition, Turnis et al. suggested that MCL-1 is required for the survival of early erythroid progenitor cells and is dispensable in later stages [46]. The extensive proliferation of immature erythrocytes in the SP and BM in this study may be the result of accelerated erythropoiesis in response to the rapid renewal of late erythrocytes and tissue hypoxia. In lymphopoiesis, early extramedullary CLP levels were suppressed, as were both intramedullary and extramedullary T, B, and NK cells; later, intramedullary CLP and B cells were suppressed, T and NK cells recovered, and extramedullary B and NK cells remained suppressed. Previous studies have shown that MCL-1 expression is important for the development and maintenance of CLPs, as well as B- and T-lymphocytes [47,48,49]. Interestingly, B-lymphocytes and NK cells showed similar drug–response profiles, with high sensitivity to the BCL-2 inhibitor Venetoclax [43].

Under the action of the drug, extramedullary hematopoiesis in the organism was significantly altered, and the production of blood cells of all lineages was affected. Among them, compensatory hematopoiesis occurred in the myeloid and megakaryocytic erythroid lineages, both of which recovered later; lymphocytes in both the BM and extramedullary organs were inhibited and recovered at a slow pace. In conclusion, the targeted small-molecule inhibitors significantly hindered hematopoiesis of the lymphoid lineage, while myeloid-lineage hematopoiesis was not greatly affected.

## 5. Conclusions

In the present study, we designed a mouse model of the MCL-1-specific small-molecule inhibitor S63845 to study the in vivo effects of the targeted drug on hematopoiesis in various lineages. The effects of the drug on myelopoiesis, megakaryopoiesis, and erythropoiesis mainly manifested as extramedullary compensatory hematopoiesis, with a higher proportion of relatively mature myeloid and differentiated Mk-lineage cells, and erythroid cells were mostly blocked in the early stage of differentiation in progenitors; the effects of this on lymphopoiesis were inhibited in both intramedullary and extramedullary hematopoiesis. This study provides insight into the response patterns of intramedullary and extramedullary hematopoiesis in response to targeted drug stimulation, and its findings may serve as a guide for the clinical design of antitumor drug combination regimens and for the prediction and prevention of drug-induced adverse hematopoiesis-related effects.

## Figures and Tables

**Figure 1 pharmaceutics-15-01085-f001:**
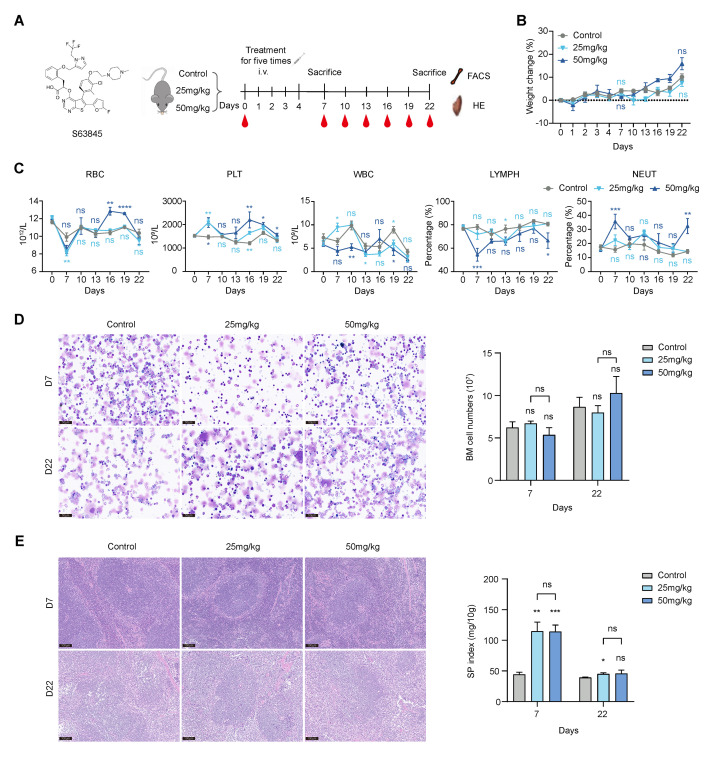
Effect of S63845 treatment on the hematopoiesis of CB6F1 mice. (**A**) Schematic representation of S63845-treated hematopoietic suppression in CB6F1 mice and the chemical structure of S63845. (**B**) Body weight change after S63845 treatment of CB6F1 mice. n ≥ 4 mice per group. (**C**) Chronological change in peripheral blood cells after S63845 treatment of CB6F1 mice. n ≥ 4 mice per group. (**D**) Representative BM Wright–Giemsa-stained smears (**left**) and complete BM counts (**right**) of control and S63845-treated mice on day 7 and day 22. Scale bars, 50 μm. Control, n = 5 (day 7), n = 6 (day 22); 25 mg/kg, n = 5 (both days 7 and 22); 50 mg/kg, n = 4 (both days 7 and 22). (**E**) Representative SP HE-stained sections (**left**) and SP index (**right**) of control and S63845-treated mice on day 7 and day 22. Scale bars, 100 μm. Control, n = 5 (day 7), n = 6 (day 22); 25 mg/kg, n = 5 (both days 7 and 22); 50 mg/kg, n = 4 (both days 7 and 22). All data represent the means ± SEM compared with control unless otherwise specified. * *p* < 0.05, ** *p* < 0.01, *** *p* < 0.001, **** *p* < 0.0001, and ns = not significant according to two-tailed unpaired Students’ *t*-test. FACS: fluorescence-activated cell sorting; HE: hematoxylin-eosin; RBC: red blood cell; PLT: platelet; WBC: white blood cell; LYMPH: lymphocyte; NEUT: neutrophil; BM: bone marrow; SP: spleen.

**Figure 2 pharmaceutics-15-01085-f002:**
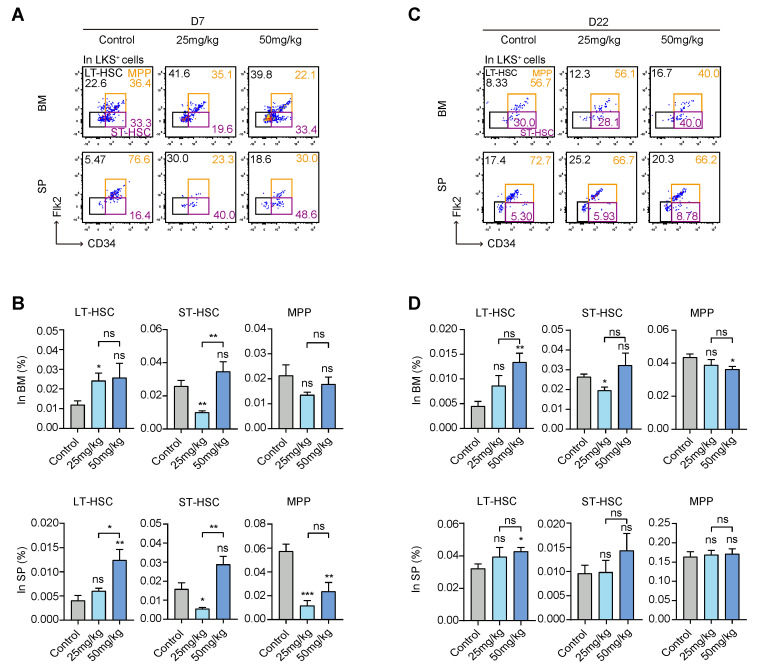
Effects of S63845 treatment on the hematopoietic stem/progenitor cells of CB6F1 mice. (**A**) Representative FACS profiles of LT-HSC, ST-HSC, and MPP subpopulations in BM and SP LKS^+^ cells of control and S63845-treated mice on day 7. Control, n = 5; 25 mg/kg, n = 5; 50 mg/kg, n = 4. (**B**) Percentage of LT-HSC, ST-HSC, and MPP populations in BM (upper panel) and SP (lower panel) LKS^+^ cells, as described in (**A**). Control, n = 5; 25 mg/kg, n = 5; 50 mg/kg, n = 4. (**C**) Representative FACS profiles of LT-HSC, ST-HSC, and MPP subpopulations in BM and SP LKS^+^ cells of control and S63845-treated mice on day 22. Control, n = 6; 25 mg/kg, n = 5; 50 mg/kg, n = 4. (**D**) Percentage of LT-HSC, ST-HSC, and MPP populations in BM (upper panel) and SP (lower panel) LKS^+^ cells, as described in (**C**). Control, n = 6; 25 mg/kg, n = 5; 50 mg/kg, n = 4. All data represent the means ± SEM compared with control unless otherwise specified. * *p* < 0.05, ** *p* < 0.01, *** *p* < 0.001, and ns = not significant according to two-tailed unpaired Students’ *t*-test. LT-HSC: long-term hematopoietic stem cell; ST-HSC: short-term hematopoietic stem cell; MPP: multipotent progenitor; LKS^+^: Lin^−^c-Kit^+^Sca-1^+^.

**Figure 3 pharmaceutics-15-01085-f003:**
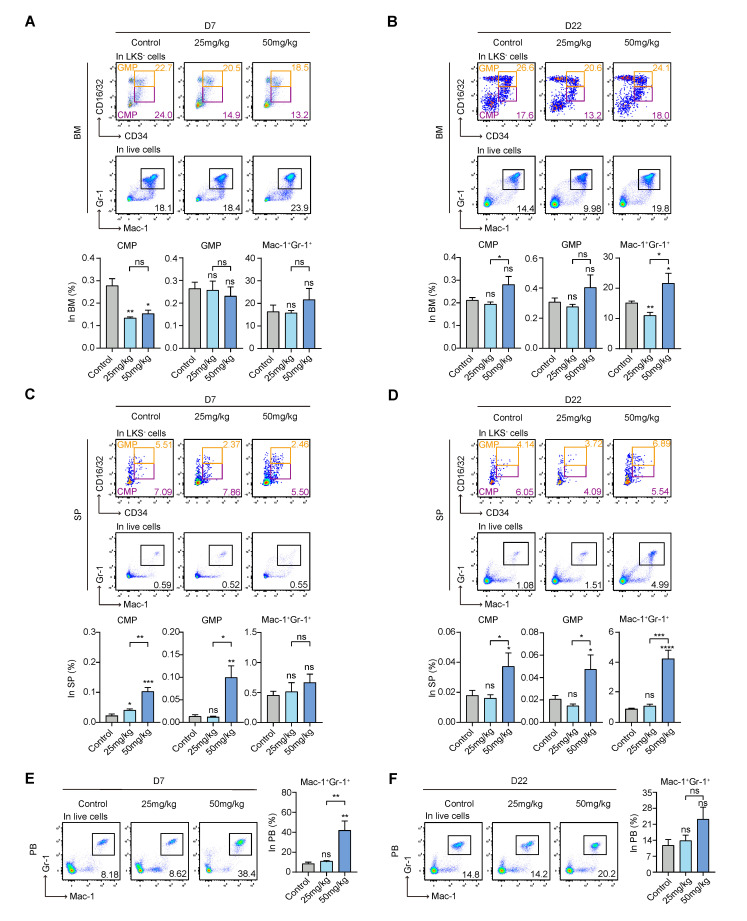
Effect of S63845 treatment on myeloid differentiation in CB6F1 mice. (**A**) Representative FACS profiles (upper panel) and percentages (lower panel) of CMP, GMP, and Mac-1^+^Gr-1^+^ subpopulations in BM cells of control and S63845-treated mice on day 7. Control, n = 5; 25 mg/kg, n = 5; 50 mg/kg, n = 4. (**B**) Representative FACS profiles and percentages of CMP, GMP, and Mac-1^+^Gr-1^+^ subpopulations in BM cells of control and S63845-treated mice on day 22. Control, n = 6; 25 mg/kg, n = 5; 50 mg/kg, n = 4. (**C**) Representative FACS profiles and percentages of CMP, GMP, and Mac-1^+^Gr-1^+^ subpopulations in SP cells of control and S63845-treated mice on day 7. Control, n = 5; 25 mg/kg, n = 5; 50 mg/kg, n = 4. (**D**) Representative FACS profiles and percentages of CMP, GMP, and Mac-1^+^Gr-1^+^ subpopulations in SP cells of control and S63845-treated mice on day 22. Control, n = 6; 25 mg/kg, n = 5; 50 mg/kg, n = 4. (**E**) Representative FACS profiles and percentages of Mac-1^+^Gr-1^+^ subpopulations in PB cells of control and S63845-treated mice on day 7. Control, n = 5; 25 mg/kg, n = 5; 50 mg/kg, n = 4. (**F**) Representative FACS profiles and percentages of Mac-1^+^Gr-1^+^ subpopulations in PB cells of control and S63845-treated mice on day 22. Control, n = 6; 25 mg/kg, n = 5; 50 mg/kg, n = 4. All data represent the means ± SEM compared with control unless otherwise specified. * *p* < 0.05, ** *p* < 0.01, *** *p* < 0.001, **** *p* < 0.0001, and ns = not significant according to two-tailed unpaired Students’ *t*-test. CMP: common myeloid progenitor; GMP: granulocyte-monocyte progenitor; LKS^−^: Lin^−^c-Kit^+^Sca-1^−^; PB: peripheral blood.

**Figure 4 pharmaceutics-15-01085-f004:**
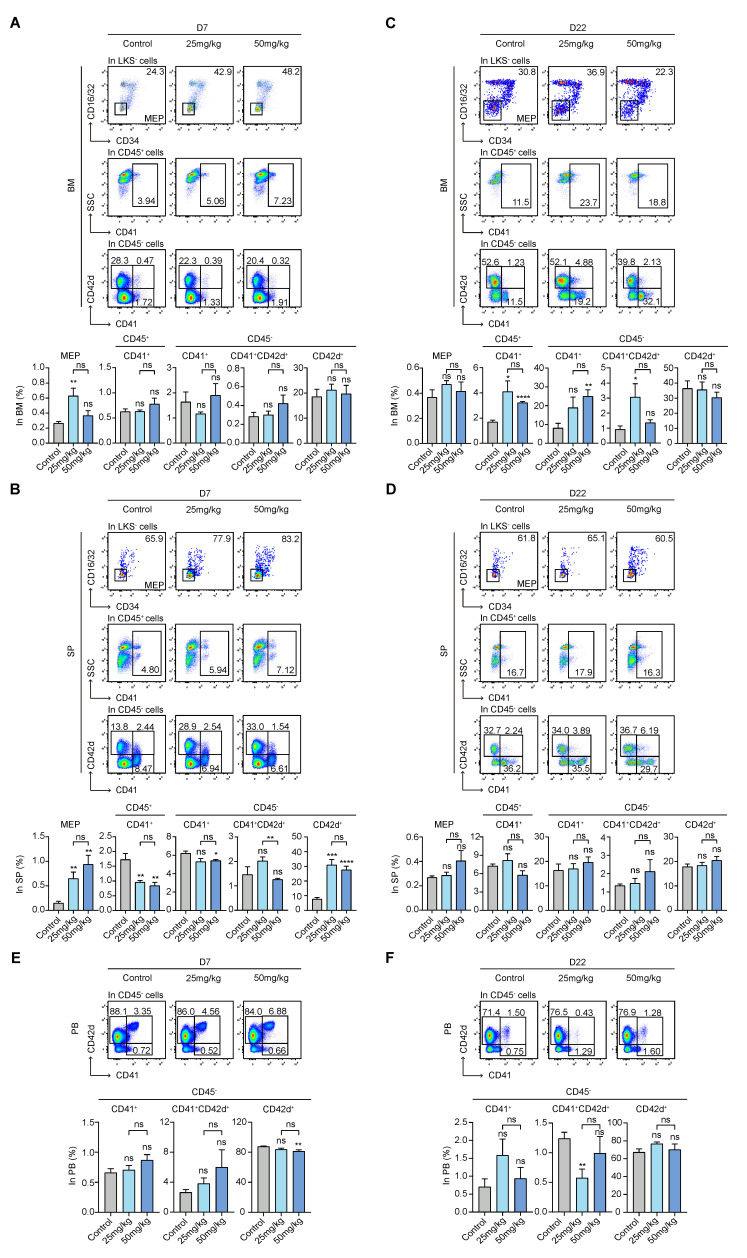
Effect of S63845 treatment on megakaryocytic differentiation in CB6F1 mice. (**A**) Representative FACS profiles and percentages of MEP, CD45^+^CD41^+^, CD45^−^CD41^+^, CD45^−^CD41^+^CD42d^+^, and CD45^−^CD42d^+^ subpopulations in BM cells of control and S63845-treated mice on day 7. Control, n = 5; 25 mg/kg, n = 5; 50 mg/kg, n = 4. (**B**) Representative FACS profiles and percentages of MEP, CD45^+^CD41^+^, CD45^−^CD41^+^, CD45^−^CD41^+^CD42d^+^, and CD45^−^CD42d^+^ subpopulations in SP cells of control and S63845-treated mice on day 7. Control, n = 5; 25 mg/kg, n = 5; 50 mg/kg, n = 4. (**C**) Representative FACS profiles and percentages of MEP, CD45^+^CD41^+^, CD45^−^CD41^+^, CD45^−^CD41^+^CD42d^+^, and CD45^−^CD42d^+^ subpopulations in BM cells of control and S63845-treated mice on day 22. Control, n = 6; 25 mg/kg, n = 5; 50 mg/kg, n = 4. (**D**) Representative FACS profiles and percentages of MEP, CD45^+^CD41^+^, CD45^−^CD41^+^, CD45^−^CD41^+^CD42d^+^, and CD45^−^CD42d^+^ subpopulations in SP cells of control and S63845-treated mice on day 22. Control, n = 6; 25 mg/kg, n = 5; 50 mg/kg, n = 4. (**E**) Representative FACS profiles and percentages of CD45^−^CD41^+^, CD45^−^CD41^+^CD42d^+^, and CD45^−^CD42d^+^ subpopulations in PB cells of control and S63845-treated mice on day 7. Control, n = 5; 25 mg/kg, n = 5; 50 mg/kg, n = 4. (**F**) Representative FACS profiles and percentages of CD45^−^CD41^+^, CD45^−^CD41^+^CD42d^+^, and CD45^−^CD42d^+^ subpopulations in PB cells of control and S63845-treated mice on day 22. Control, n = 6; 25 mg/kg, n = 5; 50 mg/kg, n = 4. All data represent the means ± SEM compared with control unless otherwise specified. * *p* < 0.05, ** *p* < 0.01, *** *p* < 0.001, **** *p* < 0.0001, and ns = not significant according to two-tailed unpaired Students’ *t*-test. MEP: megakaryocyte-erythroid progenitor.

**Figure 5 pharmaceutics-15-01085-f005:**
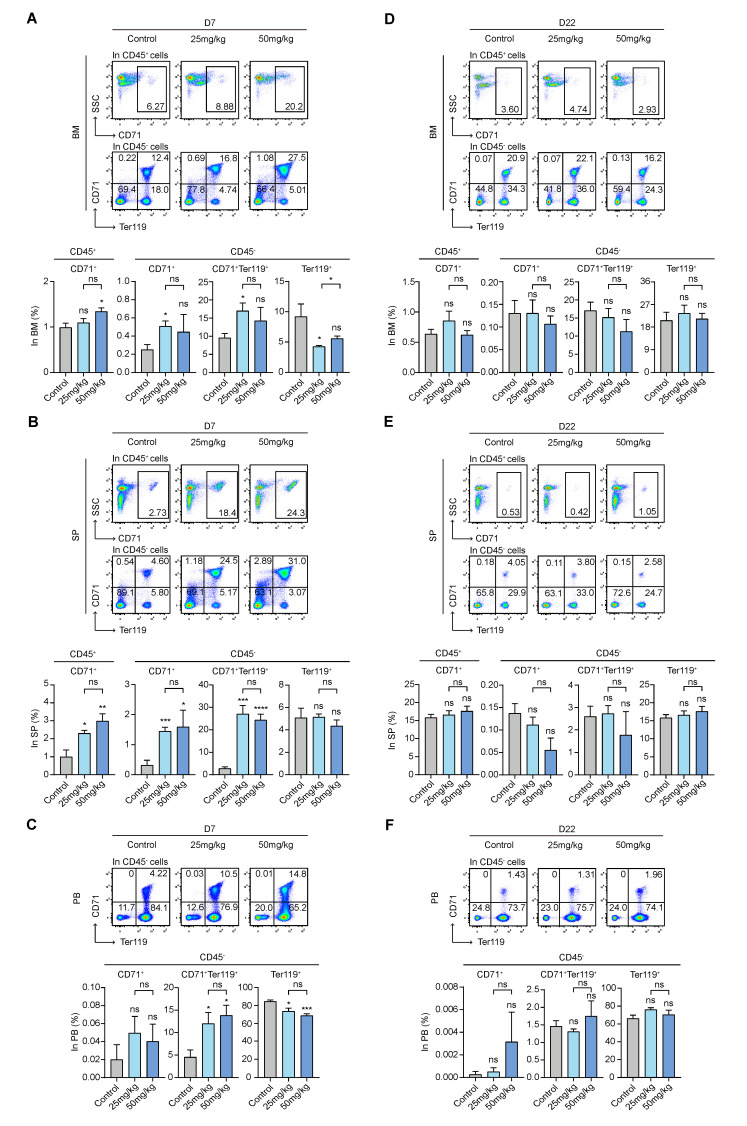
Effect of S63845 treatment on erythroid differentiation in CB6F1 mice. (**A**) Representative FACS profiles and percentages of CD45^+^CD71^+^, CD45^−^CD71^+^, CD45^−^CD71^+^Ter119^+^, and CD45^−^Ter119^+^ subpopulations in BM cells of control and S63845-treated mice on day 7. Control, n = 5; 25 mg/kg, n = 5; 50 mg/kg, n = 4. (**B**) Representative FACS profiles and percentages of CD45^+^CD71^+^, CD45^−^CD71^+^, CD45^−^CD71^+^Ter119^+^, and CD45^−^Ter119^+^ subpopulations in SP cells of control and S63845-treated mice on day 7. Control, n = 5; 25 mg/kg, n = 5; 50 mg/kg, n = 4. (**C**) Representative FACS profiles and percentages of CD45^−^CD71^+^, CD45^−^CD71^+^Ter119^+^, and CD45^−^Ter119^+^ subpopulations in PB cells of control and S63845-treated mice on day 7. Control, n = 5; 25 mg/kg, n = 5; 50 mg/kg, n = 4. (**D**) Representative FACS profiles and percentages of CD45^+^CD71^+^, CD45^−^CD71^+^, CD45^−^CD71^+^Ter119^+^, and CD45^−^Ter119^+^ subpopulations in BM cells of control and S63845-treated mice on day 22. Control, n = 6; 25 mg/kg, n = 5; 50 mg/kg, n = 4. (**E**) Representative FACS profiles and percentages of CD45^+^CD71^+^, CD45^−^CD71^+^, CD45^−^CD71^+^Ter119^+^, and CD45^−^Ter119^+^ subpopulations in SP cells of control and S63845-treated mice on day 22. Control, n = 6; 25 mg/kg, n = 5; 50 mg/kg, n = 4. (**F**) Representative FACS profiles and percentages of CD45^−^CD71^+^, CD45^−^CD71^+^Ter119^+^, and CD45^−^Ter119^+^ subpopulations in PB cells of control and S63845-treated mice on day 22. Control, n = 6; 25 mg/kg, n = 5; 50 mg/kg, n = 4. All data represent the means ± SEM compared with control unless otherwise specified. * *p* < 0.05, ** *p* < 0.01, *** *p* < 0.001, **** *p* < 0.0001, and ns = not significant according to two-tailed unpaired Students’ *t*-test.

**Figure 6 pharmaceutics-15-01085-f006:**
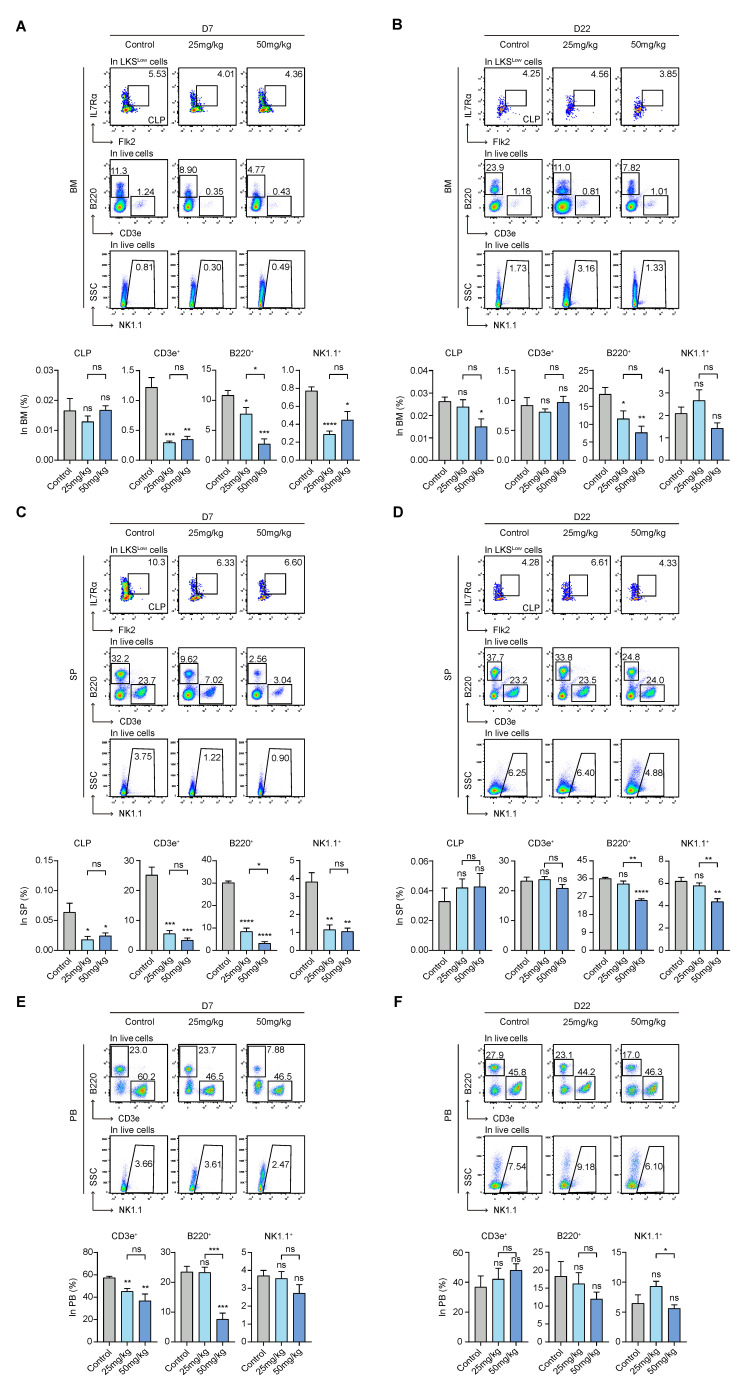
Effect of S63845 treatment on lymphoid differentiation in CB6F1 mice. (**A**) Representative FACS profiles and percentages of CLP, CD3e^+^, B220^+^, and NK1.1^+^ subpopulations in BM cells of control and S63845-treated mice on day 7. Control, n = 5; 25 mg/kg, n = 5; 50 mg/kg, n = 4. (**B**) Representative FACS profiles and percentages of CLP, CD3e^+^, B220^+^, and NK1.1^+^ subpopulations in BM cells of control and S63845-treated mice on day 22. Control, n = 6; 25 mg/kg, n = 5; 50 mg/kg, n = 4. (**C**) Representative FACS profiles and percentages of CLP, CD3e^+^, B220^+^, and NK1.1^+^ subpopulations in SP cells of control and S63845-treated mice on day 7. Control, n = 5; 25 mg/kg, n = 5; 50 mg/kg, n = 4. (**D**) Representative FACS profiles and percentages of CLP, CD3e^+^, B220^+^, and NK1.1^+^ subpopulations in SP cells of control and S63845-treated mice on day 22. Control, n = 6; 25 mg/kg, n = 5; 50 mg/kg, n = 4. (**E**) Representative FACS profiles and percentages of CD3e^+^, B220^+^, and NK1.1^+^ subpopulations in PB cells of control and S63845-treated mice on day 7. Control, n = 5; 25 mg/kg, n = 5; 50 mg/kg, n = 4. (**F**) Representative FACS profiles and percentages of CD3e^+^, B220^+^, and NK1.1^+^ subpopulations in PB cells of control and S63845-treated mice on day 22. Control, n = 6; 25 mg/kg, n = 5; 50 mg/kg, n = 4. All data represent the means ± SEM compared with control unless otherwise specified. * *p* < 0.05, ** *p* < 0.01, *** *p* < 0.001, **** *p* < 0.0001, and ns = not significant by two-tailed unpaired Students’ *t*-test. CLP: common lymphoid progenitor; LKS: Lin^−^c-Kit^Low^Sca-1^Low^.

## Data Availability

All data generated or analyzed during this study are included in this article. Any other data are available from the corresponding author upon reasonable request.

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
