# Peer review of "MCL-1 Inhibitor S63845 Distinctively Affects Intramedullary and Extramedullary Hematopoiesis"

_pharmaceutics, 2023, doi:10.3390/pharmaceutics15041085_

Round 1

Reviewer 1 Report

The article "MCL-1 inhibitor S63845 distinctively affects intramedullary and extramedullary hematopoiesis" is an interesting study about the hematopoietic injury induced by the small molecule S63845 in healthy CB6F1 mice. The article is well organized, scientifically sound and present noteworthy data in the field. Hence, it is suitable for publications after the following minor revisions:

- When describing clinical trials, add the NCT number to better identify the study. For example in lines: 49, 67 and 69.

-  Rephrase lines 95-99.

- In figure 1, add the chemical structure of S63845.

- Some typo and grammar errors must be corrected. For example, was reduced in lines 234 and 237; missing verb in lines 242-243; change Vinetoclax with Venetoclax in line 495.

- Lines 40-43 have missing citations. Add appropriate references about "small-molecule targeted pharmaceuticals": Drug Dev Res. 2022 Sep;83(6):1331-1341. doi: 10.1002/ddr.21962. Epub 2022 Jun 24. PMID: 35749723; J. Med. Chem. 2020, 63, 21, 12403–12428 https://doi.org/10.1021/acs.jmedchem.0c00696; Eur J Med Chem. 2022 Dec 5;243:114744. doi: 10.1016/j.ejmech.2022.114744; Bioorg Chem. 2021 Jul;112:104965. doi: 10.1016/j.bioorg.2021.104965.

Reviewer 2 Report

In this study, the authors investigate the effects of S63845, a small molecule inhibitor targeting Mcl1, in the hematopoietic mouse model. The authors found that Mcl1 inhibitor affect the intramedullary and extramedullary hematopoiesis. It is a nice written paper and I have several comments below.

  1. The authors should compare the effect of S63845 in the intramedullary and extramedullary hematopoiesis with other Mcl1 inhibitors such as A1210477 to test whether S63845 is a best choice among the Mcl1 inhibitors.

  2. The authors should test the effect of S63845 in the intramedullary and extramedullary hematopoiesis in mouse models with different backgrounds to validate their findings. 

  3. Why the authors choose CB6F1 mice for this type of experiment. Please explain in the material and methods

Reviewer 3 Report

MCL-1 inhibitor S63845 distinctively affects intramedullary and extramedullary hematopoiesis

They present a study of a new mouse model that they designed specifically for MCL-1-screening of the inhibitor S63845 and to assess the in vivo effects of S63845 on hematopoiesis in multiple mice. 

The result of their drug study showed compensatory hematopoiesis as a result of myelopoiesis, megakaryopoiesis and erythropoiesis within differentiated Mk-lineage cells. Looking at both intramedullary and extramedullary hematopoiesis, erythroid cells were blocked at the early stages.

Response patters were shown to be in direct correlation to the use of the drug S63845 and they purport this could be a good control for use in clinical studies where combination therapies are tested for antitumor proliferation and screening of adverse effects (preclinical studies).

Overall the data is supportive and the findings are consistent. They had good amounts of data in their pools studies.

No gender of mice is mentioned (male or female), so it is hard to determine if sex was taken into account given that omission of data in the results. They do say “randomly and evenly classified ” but that is hard to know if it is weight, age, or sex or none of the above.

Please clarify this in your text for the reader to explicitly know what random indicates with regard to the mice (gender, age, weight, etc).

Collection and screening protocols (# days and when collections occurred and how they were processed) seemed sufficiently adequate.

Overall data is useful and its publication seems important.

Round 2

Reviewer 2 Report

The authors do not performed any experiments suggested by reviewer. They just mentioned that they will conduct it in future but dont know exactly when.

I feel the quality of this manuscript is not deserved to publish in Pharmaceutics journal with current status. The authors should revise the manuscript as suggested by reviewer's comments.